# Insights on the Selection of the Coefficient of Variation to Assess Speed Fluctuation in Swimming

**DOI:** 10.3390/jfmk9030129

**Published:** 2024-07-25

**Authors:** Mafalda P. Pinto, Daniel A. Marinho, Henrique P. Neiva, Tiago M. Barbosa, Jorge E. Morais

**Affiliations:** 1Department of Sport Sciences, University of Beira Interior, 6201-001 Covilhã, Portugalmarinho.d@gmail.com (D.A.M.); henriquepn@gmail.com (H.P.N.); 2Research Centre in Sports, Health and Human Development (CIDESD), 6201-001 Covilhã, Portugal; 3Department of Sport Sciences, Instituto Politécnico de Bragança, 5300-253 Bragança, Portugal; barbosa@ipb.pt; 4Research Centre for Active Living and Wellbeing (LiveWell), Instituto Politécnico de Bragança, 5300-253 Bragança, Portugal

**Keywords:** statistical parametric mapping, continuous analysis, comparison, swimmers, performance

## Abstract

The aim of this study was to compare swimming speed and speed fluctuations in front crawl between swimmers of different performance levels using discrete variables against statistical parametric mapping (SPM). The sample was composed of 34 male swimmers divided into three groups: (i) group #1—recreational swimmers; (ii) group #2—competitive swimmers aged 12 to 14 years; (iii) group #3—competitive swimmers aged 15 to 17 years. Swimming speed and speed fluctuations (calculated based on four different conditions) were used as discrete variables. Using these discrete variables, ANOVA one-way was used to verify differences between groups, and Bonferroni post-hoc correction for pairwise comparison whenever suitable. SPM (with similar statistical tests) was used to analyze the swimming speed and fluctuation as a continuous variable. Overall, both statistical approaches revealed significant differences (*p* < 0.001) in swimming speed and speed fluctuations. However, as discrete variables (in four different conditions), the speed fluctuation was not able to detect significant differences between groups #2 and #3. Conversely, SPM was more sensitive and did yield significant differences between these two groups. Therefore, researchers and coaches should be aware that the speed fluctuation as a discrete variable may not identify differences in swimming speed fluctuations when the average value between groups is marginal. On the other hand, SPM was more sensitive in analyzing all groups.

## 1. Introduction

Swimming is characterized as being a periodic acceleration/deceleration sport [1,2]. Thus, researchers and coaches put a lot of focus on understanding the balance between thrust (acceleration) and drag (deceleration) [3,4]. From this interaction between thrust and drag, several fluctuations in swimming speed can be observed [5,6]. These fluctuations within a swim stroke cycle are usually measured by a discrete variable called the intra-cyclic variation of the horizontal velocity of the center of mass, which is a feasible way to examine the swimmers’ overall mechanics [7]. This variable is also called “speed fluctuation” and can be calculated based on: (i) the coefficient of variation (one standard deviation/mean × 100)—dv1; (ii) the difference between maximal and minimum instantaneous swimming speed—dv2; (iii) the ratio of the mean swimming speed/difference between the maximal and minimum instantaneous swimming speed—dv3; (iv) the ratio of the minimum and maximum swimming speeds/intracycle mean swimming speed—dv4 [8]. 

Both the swimming speed and speed fluctuation (this latter one irrespective of the way of calculation) are used as discrete variables, i.e., with no time dimension to understand the swimmers’ stroke kinematics [9,10,11]. In the case of swimming speed, several research groups with expertise in swimming use the average swimming speed during the intermediate section of the swimming pool [9,12,13]. Afterwards, the speed fluctuation variable (irrespective of the way) is calculated. The literature reports speed fluctuation as an indicator of swimming efficiency [7,14,15]. Indeed, the overall trend is that smaller values of speed fluctuation are related to the fastest swimming speeds [16,17]. Because it is a variable that is simple to calculate and interpret, coaches and practitioners can easily give insights to their swimmers. 

Recently, a study raised the fact that some issues could emerge when using speed fluctuation as the coefficient of variation [18]. Overall, the authors argued that researchers and coaches should take care when using the coefficient of variation as an indicator of the front crawl intra-cycle speed fluctuation since it is likely biased by the mean swimming speed. Moreover, it was suggested that analysis of the swimming speed as a continuous variable (with a time dimension) and comparing different swimming levels could bring greater practical relevance. Indeed, it was reported that using continuous analysis procedures (i.e., with a time dimension), such as statistical parametric mapping (SPM) can give deeper insights into hypothetical differences in these speed fluctuations [19,20]. This statistical method exploits the use of random field theory to perform topological inference by directly mapping the conventional Gaussian distribution onto smooth n-dimensional data [21]. By using SPM for time series data, the statistical result is still a time series (e.g., a time series of t-values) and allows for better interpretation of data [22]. SPM application is increasing in sports sciences, contributing to more detailed movement analyses in biomechanical and performance contexts [23,24,25]. In the case of swimming, for instance, it was used to investigate differences between elite and sub-elite adult swimmers in the 100 m breaststroke [20] and to identify differences within the front-crawl stroke cycle between age-group swimmers of both sexes [26]. Additionally, we could not find any information about research that used all these discrete ways of calculating speed fluctuations and comparing the outputs with a time dimension procedure such as SPM.

Therefore, the aim of this study was to compare swimming speeds in front crawl between swimmers of different performance levels using discrete variables against SPM. Based on discrete variables, both the swimming speed and speed fluctuation (calculated based on four different conditions) were compared. Based on SPM, the swimming speed was analyzed as a continuous variable, and thus all fluctuations within the stroke cycle were considered. It was hypothesized that SPM would be more sensitive in detecting differences in speed fluctuation with the add-on of identifying where in the stroke cycle such differences would occur.

## 2. Materials and Methods

### 2.1. Sample

The sample was composed of 34 male swimmers divided into three groups: #1—recreational swimmers (N = 14); #2—competitive swimmers aged 12 to 14 years (N = 10); #3—competitive swimmers aged 15 to 17 years (N = 10). Part of this sample was retrieved from the study by Morais and co-workers [12]. Their demographics are presented in Table 1. The oldest group (#1—recreational swimmers) presented the greatest body mass, height, and arm span, followed by group #3 (competitive aged 15 to 17 years), and group #2 (competitive aged 12 to 14 years), respectively. The performance level (World Aquatic Points—WAPS) of competitive swimmers was calculated based on the 100 m freestyle short-course event. They were recruited from a national team that regularly participated in regional, national, and international competitions. The sample (groups #2 and #3) included age-group national record holders, age-group national champions, and other swimmers who enrolled in national talent identification programs (Tier 3 athletes) [27]. They trained six to nine times a week. At the time of data collection, they were in peak form at the end of the second macro-cycle. As for the recreational swimmers, these were classified as Tier #1 athletes [27]. Inclusion criteria were that the swimmers should be front-crawl experts and have no limitations (e.g., no injuries in the past 6 months) that would prevent them from performing at their best. An informed consent was obtained by the coaches and/or parents and the swimmers themselves to participate in this study. All procedures followed the Declaration of Helsinki regarding human research. The Polytechnic Ethic Committee also approved the study design (N. º 72/2022). 

### 2.2. Swimming Speed and Speed Fluctuation

Swimmers were invited to perform three all-out 25 m trials, with a push-off start, with a 10-min interval to ensure full recovery. The best trial (i.e., the one with the fastest swimming speed) was used for analysis. Swimmers were instructed to perform non-breathing strokes during such a distance to avoid changes in coordination or technique [28]. Three consecutive stroke cycles between the 10th and 20th meters were analyzed. This was done to avoid any advantage of the wall push-off. The average of the three-stroke cycles was used for analysis.

The string of a mechanical apparatus (SpeedRT, ApLab, Rome, Italy) was attached to the swimmers’ waist [29]. The speedometer calculated the displacement and speed of the swimmers (*f* = 100 Hz). Afterwards, the speed–time series were imported into signal processing software (AcqKnowledge v.3.9.0, Biopac Systems, Santa Barbara, CA, USA). The signal was handled with a Butterworth 4th order low-pass filter (cut-off: 5 Hz) based on the analysis of the residual error vs. cut-off frequency output [30]. A video camera GoPro (Hero 7, San Mateo, CA, USA) filmed the swimmers in the sagittal plane to identify the hand’s water entry and exit. This was synchronized with the mechanical device. The beginning and end of each stroke cycle were set by the consecutive entry of the right hand into the water. A swim stroke cycle is composed of the following phases: (i) entry and catch; (ii) downsweep; (iii) insweep; (iv) upsweep, and; (v) exit and recovery [31]. The swimming speed (m/s) was retrieved from the software and the speed fluctuations were calculated as aforementioned (i.e., dv1, dv2, dv3, and dv4). 

### 2.3. Statistical Analysis

The mean plus one standard deviation (SD) was computed as descriptive statistics. The ANOVA one-way was used to measure differences between groups (α = 0.05). The effect size index (eta square—η^2^) was computed and interpreted as: without effect if 0 < η^2^ ≤ 0.04; minimum if 0.04 < η^2^ ≤ 0.25; moderate if 0.25 < η^2^ ≤ 0.64; strong if η^2^ > 0.64 [32]. The Bonferroni post-hoc correction was used to verify pairwise differences (*p* < 0.017). Cohen’s d estimated the standardized effect sizes and deemed as: trivial if 0 ≤ d < 0.20; small if 0.20 ≤ d < 0.60; moderate if 0.60 ≤ d < 1.20; large if 1.20 ≤ d < 2.00; very large if 2.00 ≤ d < 4.00; nearly distinct if d ≥ 4.00 [33]. 

SPM ANOVA one-way was used to verify the differences between groups (α = 0.05) [21]. SPM Bonferroni post-hoc correction was used to verify differences between pairwise (*p* < 0.017). Before such analysis, each stroke cycle was normalized to its duration on R software (version 2024.04.02). The normalization procedure implies creating a near-identical copy of each signal segment that is resampled to a normalized length of 100% [34]. In this case, all swim stroke cycles were stretched/compressed to the same length. Consequently, each normalized curve consisted of 101 points, irrespective of how many points each original curve contained. This is a procedure commonly used in human gait that allows comparison across different gait cycles, subjects, and conditions due to the inherent variability in human gait [35]. The same rationale is applied to swimming. By normalizing the stroke cycles, coaches can identify differences within the swim stroke. SPM analyses were implemented using the open source spm1d code on Matlab (v.M0.1, www.spm1d.org).

## 3. Results

Table 2 presents the descriptive data related to swimming speed and speed fluctuation by group. Swimmers in group #3 were the fastest, in group #2 the second fastest, and in group #1 the slowest. As for speed fluctuation, this was greater in group #1, followed by group #3, and group #2, respectively (all conditions). Regarding the swimming speed, the ANOVA one-way yielded significant differences between groups (F = 75.01, *p* < 0.001, η^2^ = 0.83). Bonferroni post-hoc correction revealed significant differences between group #1 and groups #2 (*p* < 0.001, d = 3.60) and #3 (*p* < 0.001, d = 4.38), but not between group #2 and #3 (Table 2). As for speed fluctuation (all conditions), these presented similar findings. That is, significant differences between groups were noted (*p* < 0.001) (Table 2). The pairwise comparison also revealed significant differences between group #1 and groups #2 and #3, but not between groups #2 and #3 (Table 1).

Figure 1 depicts the swimming speed curve differences by SPM (Panel A). There was a significant difference (F = 12.016, *p* < 0.001) between the three groups mainly over the entire stroke cycle. SPM post-hoc comparison between group #1 and #2 (Panel B) revealed significant differences mainly over the entire stroke cycle. Non-significant differences were only noted between ~16% and ~29% (end of the downsweep and insweep of the right hand), and between ~68% and ~82% (end of the downsweep and beginning of the upsweep of the left hand). The comparison between the group #1 and #3 trend (Panel C) was similar (i.e., differences over the entire stroke cycle), except between ~73% and ~80% (insweep and beginning of the upsweep of the left hand). Contrary to what was observed with the dv as a discrete variable, the post-hoc correction through SPM was able to detect significant differences between group #2 and #3 (Panel D). These were noted between ~17% and ~27% (end of the downsweep phase and insweep of the right hand), and between ~44% and ~51% (end of the downsweep phase of the right hand). 

## 4. Discussion

The aim of this study was to compare swimming speeds in front crawl between swimmers of different performance levels using discrete variables against SPM. The main findings indicate that, by analyzing discrete variables, i.e., average swimming speed and speed fluctuation (calculated based on four different conditions), significant differences between groups were noted (Table 2). However, post-hoc comparisons indicated a non-significant difference between groups #2 and #3 (similar values of speed fluctuation considering the four conditions of calculation) (Table 2). SPM was revealed to be a more sensitive analysis of the speed–time curve as a continuous time-series. Contrary to discrete variables analysis, SPM detected significant differences between groups #2 and #3 in the post-hoc correction (Figure 1, Panel D). 

Speed fluctuation (calculated based on the aforementioned four different conditions, but particularly based on the coefficient of variation) has been considered a good proxy of the intra-cyclic variation of the horizontal speed of the center of mass within a stroke cycle, and thus a mechanical efficiency indicator [7]. Among the swimming community, this is deemed to be a feasible and straightforward procedure to analyze the swimmers’ overall stroke kinematics. Moreover, several practical advantages for researchers and coaches can be listed: (i) identification of key moments in different phases of the cycle; collection of relevant details; (ii) easily interpretable data for coaches and practitioners, and; (iii) a straightforward way of setting the swimmer’s competitive level [7]. At least in front crawl, it was shown that the fastest or most highly skilled swimmers have smaller speed fluctuations (particularly calculated as the coefficient of variation and reporting to the average of the stroke cycle) than their slower or less skilled counterparts [9,10,16]. It must be mentioned that such smaller speed fluctuations might not be significant [9]. Nonetheless, this confirms that speed fluctuation can be useful in discriminating against the best or poorest swimmers and delivering important insights to coaches. 

Notwithstanding, and as mentioned previously, concerns were raised about the use of speed fluctuation (calculated as being the coefficient of variation) because this reports the average of what happens within the stroke cycle [18]. Indeed, authors who aimed to compare the swimming speed and speed fluctuation in front crawl between competitive swimmers of different age-groups reported contradictory findings [12]. Even though significant differences were noted in swimming speed between groups, speed fluctuation revealed non-significant differences. Conversely, the same comparison selecting SPM did yield significant differences in the speed fluctuation at a given moment of the stroke cycle [12]. Thus, it seems that whenever marginal differences are noted by the speed fluctuation (in such cases as being the coefficient of variation), classical statistics using discrete values are not sensitive enough to identify hypothetical differences. On the other hand, in the present study, classical statistics detected significant differences between groups (ANOVA) by both the swimming speed and speed fluctuation. One can speculate that this occurred because performance-level group #1 presented both a substantially slower swimming speed and larger speed fluctuation. But pairwise comparison revealed only significant differences between group #1 and groups #2 and #3. Once again, classical statistics did not detect significant differences between the fastest groups (#2 and #3) in swimming speed and speed fluctuation. 

In another study that compared swimmers of different performance levels, the authors noted that the poorest performance level group was significantly slower than the intermediate and fastest groups (the intermediate and fastest groups were not significantly different) [10]. However, regarding speed fluctuation, this was only significantly different between the slowest and intermediate groups with the fastest group (the slowest and intermediate groups were not significantly different) [10]. This shows that there are studies where significant differences in swimming speed were noted, but this did not happen in speed fluctuation [9,36]. This allows us to indicate that the fastest swimmers may present smaller speed fluctuations than their slower counterparts, but a cause–effect phenomenon may not happen (i.e., small speed fluctuation leads to the fastest swimming speeds). It argued that despite speed fluctuation being a gross propulsive efficiency proxy (for the same hydrodynamic drag condition), swimmers can adopt different mechanical strategies that will affect such gross swimming efficiency [37]. Aiming to better understand this swimming speed–speed fluctuation relationship, Pinto et al. [38] measured these variables in a stroke-by-stroke analysis. These authors noted that non-significant relationships were verified between the swimming speed and speed fluctuations. Moreover, this relationship (despite being non-significant) was not always inverse, i.e., small speed fluctuations did not always lead to the fastest swimming speeds [38]. This highlights the rationale indicating that this variable may not be a cause of a given behavior, but a consequence [37]. 

Notwithstanding, SPM being, a continuous time-series analysis, was more sensitive to the speed fluctuations enabling the detection of differences within the stroke cycle and pinpointing where these happened. Therefore, one can suggest that for those who aim to identify differences in speed fluctuations (where the average values are close), a continuous time-series analysis seems to be the best approach at least in comparison to classical statistics that use discrete variables. Researchers aimed to compare swimming speed and speed fluctuations at different competitive levels of young swimmers of both sexes as a discrete variable and based on SPM [26]. The main outcomes were that, overall, non-significant differences were noted between performance levels in boys, and between boys and girls of the same performance level. Significant differences were noted in girls between the two fastest performance levels and the slowest one. Which, curiously, was where the greatest absolute difference was noted [26]. Data from the present study revealed a similar trend where differences in the speed fluctuation between the two competitive groups (fastest ones) were only noted through SPM. Once again, classical statistics were able to detect differences between group #1 (recreational swimmers with the poorest performances) and the two remaining groups (competitive swimmers with the best performances). This indicates that the speed fluctuation, irrespective of the way of calculation, can only be compared based on discrete values (through classical statistics) when the groups do not present very close or similar values. For instance, in a study by Lopes and co-workers [39], the authors aimed to detect differences between two sections of the swimming pool (10–15 m vs. 15–20 m) in a set of variables related to stroke kinematics including the speed fluctuation (calculated as being the coefficient of variation). They noted significant differences in the speed fluctuation between sections (10–15 m: speed fluctuation = 33.82 ± 15.22%; 15–20 m: speed fluctuation = 25.59 ± 13.15%; *p* = 0.005; d = 0.58). Referring again to the study of Figueiredo et al. [10] it was noted that significant differences were only verified between groups that had the greatest difference. Once again, it was not possible to detect differences between groups that had close values of speed fluctuation through discrete variables. These findings corroborate our previous statement, where differences in speed fluctuation (based on discrete values) can only be detected when the average values are not close or similar. Otherwise, when the discrete values are close or similar, the continuous analysis approach (such as SPM) seems to be the most appropriate way of detecting hypothetical differences as shown in other studies [12]. 

In summary, continuous speed–time analysis by SPM analysis revealed itself to be more sensitive than the speed fluctuation (calculated based on four different conditions) when examining speed fluctuations. This procedure also has the advantage of identifying where within the stroke cycle such differences occur. At least in competitive swimming, the speed fluctuation as a discrete variable was not able to detect differences between performance groups where the average value was similar. Researchers and coaches should be aware that when the speed fluctuation values are similar between different groups, skill levels or tiers, it may not be possible to detect differences based on discrete variables. As a main limitation, it can be considered the sample size. An a priori power analysis was performed using G*Power [40]. A total of 66 participants were required to detect a large effect size (f^2^ = 0.40) with 80% power (α= 0.05) for an “ANOVA: Fixed effects, omnibus, one-way” statistical test. Therefore, researchers could aim to better understand this speed fluctuation phenomenon in different competitive levels and female swimmers as it is of paramount importance for coaches and practitioners. It is also suggested that larger sample sizes be used to understand if the outcomes are like those presented in this study. 

## 5. Conclusions

Swimming speed and speed fluctuation (calculated based on four different conditions), as discrete variables, revealed an overall significant difference between groups. However, pairwise comparison did not identify significant differences between the two fastest groups. Conversely, SPM (a continuous time-series procedure, i.e., with a time dimension) did identify significant differences between such groups. This indicates that SPM is a more sensitive approach to the analysis of swimming speed fluctuations, particularly when the differences between these are marginal. 

## Figures and Tables

**Figure 1 jfmk-09-00129-f001:**
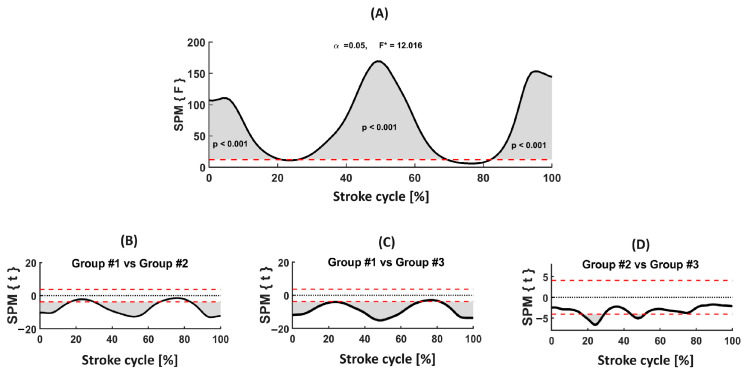
(**A**) ANOVA one-way of swimming speed by SPM between the three groups, and the corresponding post-hoc analysis (**B**)—group #1 vs. #2; (**C**)—group #1 vs. #3; (**D**)—group #2 vs. #3). {F}—variance statistic for statistical parametric mapping. SPM {t}—post-hoc statistic for statistical parametric mapping. Grey areas indicate significant differences. In panels (**B**–**D**) these areas correspond to *p* < 0.017. The dotted black line indicates the null hypothesis. Dash red lines represent the 95% confidence intervals (95 CI).

**Table 1 jfmk-09-00129-t001:** Descriptive statistics (mean ± one standard deviation—SD) of the swimmers’ demographics by group.

	Mean ± SD		
	Group #1 (N = 14)	Group #2 (N = 10)	Group #3 (N = 10)
Age [years]	20.07 ± 1.93	13.20 ± 0.79	16.39 ± 0.69
Body mass [kg]	73.87 ± 8.00	57.30 ± 8.20	70.38 ± 5.97
Height [cm]	179.48 ± 6.54	169.95 ± 8.78	177.30 ± 5.60
Arm span [cm]	183.86 ± 8.06	174.05 ± 10.08	183.60 ± 10.01
WAPS [100 m freestyle]	-	357.90 ± 47.69	578.40 ± 57.49

WAPS—World Aquatic Points.

**Table 2 jfmk-09-00129-t002:** Descriptive statistics (mean ± one standard deviation—SD) of the participants’ swimming speed and speed fluctuation (dv) as being the coefficient of variation (CV). The one-way ANOVA and pairwise comparison is also presented.

	Mean ± SD			ANOVA One-Way		Post-Hoc Comparison		
	Group #1	Group #2	Group #3	F-Ratio (*p*)	η^2^	#1 vs. #2	#1 vs. #3	#2 vs. #3
Speed [m/s] ^a,b^	1.17 ± 0.15	1.56 ± 0.03	1.67 ± 0.06	75.01 (*p* < 0.001)	0.83	*p* < 0.001; d = 3.60	*p* < 0.001; d = 4.38	-
dv1 [%] ^a,b^	23.90 ± 5.06	7.33 ± 2.11	7.69 ± 1.37	88.04 (*p* < 0.001)	0.85	*p* < 0.001; d = 4.27	*p* < 0.001; d = 4.37	-
dv2 [m/s]	0.96 ± 0.21	0.39 ± 0.11	0.45 ± 0.06	54.42 (*p* < 0.001)	0.78	*p* < 0.001; d = 3.40	*p* < 0.001; d = 3.30	-
dv3 [a.u.]	1.27 ± 0.28	4.25 ± 1.34	3.79 ± 0.53	49.47 (*p* < 0.001)	0.76	*p* < 0.001; d = 3.08	*p* < 0.001; d = 5.95	-
dv4 [a.u.]	0.36 ± 0.09	0.50 ± 0.04	0.45 ± 0.02	17.40 (*p* < 0.001)	0.53	*p* < 0.001; d = 2.01	*p* = 0.001; d = 1.38	-

dv1—speed fluctuation based on the coefficient of variation; dv2—speed fluctuation based on the difference between maximal and minimum instantaneous speed; dv3—speed fluctuation based on the ratio of the mean speed/difference between the maximal and minimum instantaneous speed; dv4—speed fluctuation based on the ratio of the minimum and maximum speeds/intracycle mean speed; η^2^—effect size index; a—significant differences (*p* < 0.001) between group #1 and #2; b—significant differences (*p* < 0.001) between group #1 and #3; d—Cohen’s d (effect size index).

## Data Availability

The data presented in this study are available on request from the corresponding author. The data are not publicly available due to privacy or ethical restrictions.

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
