# Peer review of "Insights on the Selection of the Coefficient of Variation to Assess Speed Fluctuation in Swimming"

_jfmk, 2024, doi:10.3390/jfmk9030129_

Round 1

Reviewer 1 Report

Comments and Suggestions for Authors

Dear Authors,

Congratulations on a very interesting and topical paper, especially from a methodological viewpoint. This paper produces novel findings and will spark many to incorporate SPM into their analyses. 

I would like to offer some suggestions that I believe will improve the paper or to enhance the clarity:

1. In the methods, were the 25m sprints performed from a push or a dive?

2. How was the best trial assessed to be the best?

3. What was the rationale for averaging the velocity across 3 stroke cycles?

4. Was the video signal and speedometer sychronised? How confident were you in identified the stroke cycles if this was not sycnhronised?

5. Also in the methods, I suggest you either include a power calculation (even retrospectively) or include a narrative in relation to this. 

6. A few errors in Table 1. Confirm was WAPS is (I suspect World Aquatics points). Fix the WAPS title. Would be useful to refer to Ruiz Navarro's swimming categories. 

7. In the results section (page 2). Line 88-90: were these statistically different? You can only comment on variables being different if a statistical test has been conducted.

8. Page 3, Line 98: 'dv was greatest in group 3'; but this is not reflected in Table 2. 

9. Line 103-104, does not make sense. 

10. Table 2 description includes Cohen's d, but these outputs are not included in the table.

11. Page 3, Line 112-113- refer to Panel B; Line 116-117- refer to panel C; Line 120 - refer to panel D.

12. Page 3. Line 117-122: I wonder if the upsweep/downsweep terms are suitable when they have not been defined. Consider defining the stroke cycle, and decide if the 'sweep' actions are more appropriate with respect to other terms such as 'pull' and 'push'.

13. In relation to the above point. How confident that it is the sweep actions that are accounting for the differences when the leg kick, body roll or body position between groups- can you be certain that the differences occur due to the sweeping actions- I'm not sure or convinced you can, but interested to hear your thoughts.  

14. Discussion - although this is mostly a methods piece, it would be interesting to expand on the meaning of the differences found. Is there a possibility to extrapolate the findings more? 

Comments on the Quality of English Language

English is good with some edits required to make it more scientific rather than conversational. 

Author Response

REVIEWER #1

Dear Authors,

Congratulations on a very interesting and topical paper, especially from a methodological viewpoint. This paper produces novel findings and will spark many to incorporate SPM into their analyses.

Authors: Thank you very much for the time you spent and your constructive feedback on this manuscript. We have made every effort to take on board your recommendations and comments. We hope this 2nd revised version and the responses to the comments (kindly refer to our replies below) will meet your requirements. Please note that all new changes in the revised manuscript are highlighted in yellow.

I would like to offer some suggestions that I believe will improve the paper or to enhance the clarity:

  1. In the methods, were the 25m sprints performed from a push or a dive?

Authors: We apologize for missing the info. This was from a push-off start. Added as advised.

  1. How was the best trial assessed to be the best?

Authors: The fastest one (the one with the fastest speed) was used. This was clarified in the text as advised.

  1. What was the rationale for averaging the velocity across 3 stroke cycles?

Authors: This was done to have a better representation of the swimmers’ kinematic profile.

  1. Was the video signal and speedometer synchronized? How confident were you in identifying the stroke cycles if this was not synchronized?

Authors: Both devices were synchronized. We apologize for missing this in the previous version. This was now added as advised.

  1. Also in the methods, I suggest you either include a power calculation (even retrospectively) or include a narrative in relation to this.

Authors: This was added in the limitation section due to the small sample size.

  1. A few errors in Table 1. Confirm was WAPS is (I suspect World Aquatics points). Fix the WAPS title. Would be useful to refer to Ruiz Navarro's swimming categories.

Authors: This was a formatting issue. Cleared as advised. We understand the pertinence of Ruiz-Navarro’s article, but in this case, we are not comparing swimmers from different nations. Moreover, the WAPS were only used to state the difference between groups. 

  1. In the results section (page 2). Line 88-90: were these statistically different? You can only comment on variables being different if a statistical test has been conducted.

Authors: We understand and agree with the reviewer. We changed this part to the sample characterization paragraph.

  1. Page 3, Line 98: 'dv was greatest in group 3'; but this is not reflected in Table 2.

Authors: We apologize for the typo and appreciate the reviewer’s comment. This was edited for clarity’s sake.

  1. Line 103-104, does not make sense.

Authors: These were edited for clarity.

  1. Table 2 description includes Cohen's d, but these outputs are not included in the table.

Authors: Probably formatting issues. These are now highlighted.

  1. Page 3, Line 112-113- refer to Panel B; Line 116-117- refer to panel C; Line 120 - refer to panel D.

Authors: We apologize for missing this. It was added as advised.

  1. Page 3. Line 117-122: I wonder if the upsweep/downsweep terms are suitable when they have not been defined. Consider defining the stroke cycle, and decide if the 'sweep' actions are more appropriate with respect to other terms such as 'pull' and 'push'.

Authors: The stroke cycle was defined in the methods section as advised. We appreciate the advice. We used the “sweep” terms rather than the “push and pull” which are more used in topics related to propulsion. Here we are reporting to motion.

  1. In relation to the above point. How confident that it is the sweep actions that are accounting for the differences when the leg kick, body roll or body position between groups- can you be certain that the differences occur due to the sweeping actions- I'm not sure or convinced you can, but interested to hear your thoughts.

Authors: We understand the reviewer’s comment. However, please note that we are not stating that the differences are due to such phases. We are just pointing out that are during these phases. Obviously, during upper-limbs actions, leg kicking is also being performed and body (as the reviewer mentioned) may also play a key-role. However, and once again, we are not attributing the responsibility of the differences to such actions. We are just mentioning where within the stroke cycle these happen. Indeed, and as the reviewer asked before, we defined the stroke cycle phases which are based on the upper limbs motion.

  1. Discussion - although this is mostly a methods piece, it would be interesting to expand on the meaning of the differences found. Is there a possibility to extrapolate the findings more?

Authors: We appreciate the reviewer’s advice. We elaborated a bit more in the intro and discussion sections.

Reviewer 2 Report

Comments and Suggestions for Authors

Dear corresponding author,

Thank you for submitting your article and my congratulations on your work.

Brief summary:

This study offers a significant contribution to the swimming field and also performance analysis, comparing the use of discrete variables with continuous analysis through Statistical Parametric Mapping (SPM) to evaluate both swimming speed and speed fluctuations in swimmers of different levels. The sample includes 34 male swimmers divided into three groups of different competitive levels. You used one-way ANOVA and SPM to analyze the differences between the groups. The work is well structured, methodologically sound and on my opinion provides results of great interest to researchers and coaches.

Following the journal’s rules, here are my details

General comments:

1.      The methodology is described in detail, and the use of SPM provides a more sensitive analysis compared to discrete variables. However, in my opinion the article could benefit from further clarification regarding the practical implications of the results for coaches and researchers.

2.      It would be helpful to include a more in-depth discussion of the study's limitations and possible future research directions. I hope you agree with my personal opinion.

3.      In the Introduction section, it might be useful to briefly explain how SPM analysis works for readers less familiar with this technique. This would help better understand this approach's advantages compared to traditional analysis.

4.      In the Methods section, it would be interesting to include more details on the procedure for normalizing stroke cycles before SPM analysis. How was the variability in cycle duration between participants managed?

5.      In the Results, Figure 1 is very informative, but could benefit from some modifications to improve readability. For example, you could consider using different colours for the lines of different groups and adding clearer labels for the swimming phases.

The study is aldo adequately conceived and executed, with a clear research hypothesis and adequate methodology. The innovative approach of comparing traditional analysis with discrete variables and continuous SPM analysis is particularly appreciable. The results appear to demonstrate the greater sensitivity of SPM analysis in detecting differences between groups of swimmers, especially when mean values are similar.

However, there are some points that could be improved to further strengthen the manuscript:

Specific comments:

·         Line 45: It might be useful to specify the exact number of swimmers in each group in the introductory paragraph.

·         Line 54: you write that the swimmers are of "competitive" level but it's not clear what level you consider competitive, I think is necessary some quantitative data. Let me explain better, since the swimmer's drag is directly related to their technique, it would be appropriate to define the real level of the athletes based on international Swimming Federation scores, in the absence of this it's a strong criticism.

·         Lines 62-63: "Three consecutive stroke cycles between the 10th and 20th meters were analyzed." It would be useful to briefly explain why this specific interval was chosen, it can be an interesting approach but I don't understand on what consistent scientific reference it was decided to adopt it. Maybe it is my limit, please explain better to me.

·         Line 67: Provide more details on signal analysis and Butterworth filtering, some things seem unclear.

·         Line 72: In the description of the dv calculation, it might be useful to provide the exact formula used for the coefficient of variation, for greater clarity in this case as well.

·         Lines 84-85: "Before such analysis, each stroke cycle was normalized to its duration on R software." This step is crucial and would deserve a more detailed explanation of the normalization procedure. I apologize for my excessive request but the study is interesting even in the details and I believe that some aspects should be clarified further.

·         Lines 149-152: "Indeed, a study that aimed to compare the swimming speed and dv in front-crawl between competitive swimmers of different age-groups reported contradictory findings [12]." This is an important observation that could be further elaborated to emphasize the relevance of the present study.

·         Lines 185-189: The conclusions are adequatly formulated, but it might be useful to add a brief discussion on the practical implications of these results for coaches and researchers. Although the study is difficult to transfer to the field, it may be useful to attempt this approach.

·         Figure 1: Consider adding a more detailed legend to facilitate the interpretation of the results.

In general, this is an excellent study that provides valuable insights into swimming performance analysis. With some minor revisions, I believe this article will be a significant contribution to the literature in the field. I congratulate the authors again on their work and encourage them to consider these suggestions to further improve the manuscript.

For my part, the work is certainly publishable but i have some doubts that could come to a reader who has great experience in swimming but limited in this approach to SPM, I sincerely believe that a reflection on the points i have indicated would be of useful interest to the reader.

I remain waiting and curious to await the final version as soon as possible!

Author Response

REVIEWER #2

Dear corresponding author,

Thank you for submitting your article and my congratulations on your work.

Brief summary:

This study offers a significant contribution to the swimming field and also performance analysis, comparing the use of discrete variables with continuous analysis through Statistical Parametric Mapping (SPM) to evaluate both swimming speed and speed fluctuations in swimmers of different levels. The sample includes 34 male swimmers divided into three groups of different competitive levels. You used one-way ANOVA and SPM to analyze the differences between the groups. The work is well structured, methodologically sound and on my opinion provides results of great interest to researchers and coaches.

Authors: Thank you very much for the time you spent and your constructive feedback on this manuscript. We have made every effort to take on board your recommendations and comments. We hope this 2nd revised version and the responses to the comments (kindly refer to our replies below) will meet your requirements. Please note that all new changes in the revised manuscript are highlighted in yellow.

Following the journal’s rules, here are my details

General comments:

  1. The methodology is described in detail, and the use of SPM provides a more sensitive analysis compared to discrete variables. However, in my opinion the article could benefit from further clarification regarding the practical implications of the results for coaches and researchers.

Authors: We understand and agree with the reviewer. Thus, we elaborated a bit more in the intro and discussion sections.

  1. It would be helpful to include a more in-depth discussion of the study's limitations and possible future research directions. I hope you agree with my personal opinion.

Authors: We appreciate the reviewer’s advice. These were added as advised.

  1. In the Introduction section, it might be useful to briefly explain how SPM analysis works for readers less familiar with this technique. This would help better understand this approach's advantages compared to traditional analysis.

Authors: We appreciate the reviewer’s advice. This was added as advised.

  1. In the Methods section, it would be interesting to include more details on the procedure for normalizing stroke cycles before SPM analysis. How was the variability in cycle duration between participants managed?

Authors: This info was added as suggested. Please note that the variability in cycle duration can be measured through the coefficient of variation which is one way to measure it’s fluctuation (in this case swim speed fluctuation). This is why cycle normalization (i.e., 0 to 100%) is needed to compare this based on continuous analysis.

  1. In the Results, Figure 1 is very informative, but could benefit from some modifications to improve readability. For example, you could consider using different colours for the lines of different groups and adding clearer labels for the swimming phases.

Authors: We appreciate the reviewer’s advice. This was done for clarity’s sake.

The study is also adequately conceived and executed, with a clear research hypothesis and adequate methodology. The innovative approach of comparing traditional analysis with discrete variables and continuous SPM analysis is particularly appreciable. The results appear to demonstrate the greater sensitivity of SPM analysis in detecting differences between groups of swimmers, especially when mean values are similar.

However, there are some points that could be improved to further strengthen the manuscript:

Specific comments:

  • Line 45: It might be useful to specify the exact number of swimmers in each group in the introductory paragraph.

Authors: Please note that this is immediately aften the sample section.

  • Line 54: you write that the swimmers are of "competitive" level but it's not clear what level you consider competitive, I think is necessary some quantitative data. Let me explain better, since the swimmer's drag is directly related to their technique, it would be appropriate to define the real level of the athletes based on international Swimming Federation scores, in the absence of this it's a strong criticism.

Authors: We understand the reviewer’s comment. We added information about this topic. Swimmers are characterized by their WAPS (World Aquatic Points) and we added the level categorization based on the McKay guidelines (McKay, A. K., Stellingwerff, T., Smith, E. S., Martin, D. T., Mujika, I., Goosey-Tolfrey, V. L., ... & Burke, L. M. (2021). Defining training and performance caliber: a participant classification framework. International journal of sports physiology and performance, 17(2), 317-331).

  • Lines 62-63: "Three consecutive stroke cycles between the 10th and 20th meters were analyzed." It would be useful to briefly explain why this specific interval was chosen, it can be an interesting approach but I don't understand on what consistent scientific reference it was decided to adopt it. Maybe it is my limit, please explain better to me.

Authors: We understand the reviewer’s comment. This was done to avoid the wall push-off performed by the swimmers. After the 10th meter mark, swimmers are already animated by their so called “clean swim”, i.e., without any external advantage (start by the block or wall push-off). That is, researchers measure the swimmers swim speed in the intermediate section of the swimming pool. This is a standard protocol in both experimental (Silva, A. F., Figueiredo, P., Ribeiro, J., Alves, F., Vilas-Boas, J. P., Seifert, L., & Fernandes, R. J. (2019). Integrated analysis of young swimmers’ sprint performance. Motor control, 23(3), 354-364; Barbosa, T. M., Bartolomeu, R., Morais, J. E., & Costa, M. J. (2019). Skillful swimming in age-groups is determined by anthropometrics, biomechanics and energetics. Frontiers in physiology, 10, 73) and observational studies in race analysis (Morais, J. E., Marinho, D. A., Arellano, R., & Barbosa, T. M. (2019). Start and turn performances of elite sprinters at the 2016 European Championships in swimming. Sports biomechanics, 18(1), 100-114).

In this section, swimmers may perform more complete stroke cycles, but maybe not all of them can perform five or six. Thus, to standardize this (i.e., use the same number of cycles in all swimmers), we are sure that all swimmers complete three stroke cycles. As mentioned above, this is a standard procedure in swimming studies including our research group.

  • Line 67: Provide more details on signal analysis and Butterworth filtering, some things seem unclear.

Authors: We made minor adjustments and added a reference for clarity’s sake. Please note that this is a standard procedure whenever dealing with signals including swimming (Figueiredo, P., Silva, A., Sampaio, A., Vilas-Boas, J. P., & Fernandes, R. J. (2016). Front crawl sprint performance: A cluster analysis of biomechanics, energetics, coordinative, and anthropometric determinants in young swimmers. Motor control, 20(3), 209-221; Silva, A. F., Figueiredo, P., Ribeiro, J., Alves, F., Vilas-Boas, J. P., Seifert, L., & Fernandes, R. J. (2019). Integrated analysis of young swimmers’ sprint performance. Motor control, 23(3), 354-364; Barbosa, T. M., Bartolomeu, R., Morais, J. E., & Costa, M. J. (2019). Skillful swimming in age-groups is determined by anthropometrics, biomechanics and energetics. Frontiers in physiology, 10, 73).

  • Line 72: In the description of the dv calculation, it might be useful to provide the exact formula used for the coefficient of variation, for greater clarity in this case as well.

Authors: Please note that this is now explained in the intro section.

  • Lines 84-85: "Before such analysis, each stroke cycle was normalized to its duration on R software." This step is crucial and would deserve a more detailed explanation of the normalization procedure. I apologize for my excessive request but the study is interesting even in the details and I believe that some aspects should be clarified further.

Authors: We understand, agree, and appreciate the reviewer’s comment. We elaborated a bit more as advised.

  • Lines 149-152: "Indeed, a study that aimed to compare the swimming speed and dv in front-crawl between competitive swimmers of different age-groups reported contradictory findings [12]." This is an important observation that could be further elaborated to emphasize the relevance of the present study.

Authors: We elaborated about this topic as advised.

  • Lines 185-189: The conclusions are adequatly formulated, but it might be useful to add a brief discussion on the practical implications of these results for coaches and researchers. Although the study is difficult to transfer to the field, it may be useful to attempt this approach.

Authors: We appreciate the reviewer’s comment. We elaborated a bit more as advised.

  • Figure 1: Consider adding a more detailed legend to facilitate the interpretation of the results.

Authors: Done as advised.

In general, this is an excellent study that provides valuable insights into swimming performance analysis. With some minor revisions, I believe this article will be a significant contribution to the literature in the field. I congratulate the authors again on their work and encourage them to consider these suggestions to further improve the manuscript. For my part, the work is certainly publishable but i have some doubts that could come to a reader who has great experience in swimming but limited in this approach to SPM, I sincerely believe that a reflection on the points i have indicated would be of useful interest to the reader.

I remain waiting and curious to await the final version as soon as possible!

Authors: We really appreciate the reviewer’s constructive feedback.

Reviewer 3 Report

Comments and Suggestions for Authors

General comments

Thank you for the opportunity in reviewing this study, it is a nice well written study with interesting findings which I think only requires minor changes to enhance the work.

I do think something needs adding to the introduction around how dv impacts on overall performance? This would strengthen the rational of the study.

Specific comments

Abstract

L19 – Please can you define what differences were observed.

Introduction

L28 – Please change “accelerate/decelerate” and write in full prose e.g. “acceleration or deceleration”.

Methods

L51 – Define each group based on McKay

McKay AKA, Stellingwerff T, Smith ES, Martin DT, Mujika I, Goosey-Tolfrey VL, Sheppard J, Burke LM. Defining Training and Performance Caliber: A Participant Classification Framework. Int J Sports Physiol Perform. 2022 Feb 1;17(2):317-331. doi: 10.1123/ijspp.2021-0451. Epub 2022 Dec 29. PMID: 34965513.

L60 – Could rework the short sentence here in to the previous sentence.

L62-63 – For readers interpretation, could you identify an average of how many strokes to cover the 25m? This would give a better indication of the relative proportion 3 strokes have on the swim.

L64 – Has anyone identified the reliability of the SpeedRT? Can you reference a source or any pilot data you have here?

Discussion

L152 – A study cannot aim it is inanimate, please change accordingly.

L171 – Please add the reference after Lopes for an in-text citation.

Comments on the Quality of English Language

No comment

Author Response

REVIEWER #3

General comments

Thank you for the opportunity in reviewing this study, it is a nice well written study with interesting findings which I think only requires minor changes to enhance the work. I do think something needs adding to the introduction around how dv impacts on overall performance? This would strengthen the rational of the study.

Authors: Thank you very much for the time you spent and your constructive feedback on this manuscript. We have made every effort to take on board your recommendations and comments. We hope this 2nd revised version and the responses to the comments (kindly refer to our replies below) will meet your requirements. Please note that all new changes in the revised manuscript are highlighted in yellow.

Specific comments

Abstract

L19 – Please can you define what differences were observed.

Authors: Edited for clarity’s sake.

Introduction

L28 – Please change “accelerate/decelerate” and write in full prose e.g. “acceleration or deceleration”.

Authors: We appreciate the reviewer’s advice. Edited as advised.

Methods

L51 – Define each group based on McKay

McKay AKA, Stellingwerff T, Smith ES, Martin DT, Mujika I, Goosey-Tolfrey VL, Sheppard J, Burke LM. Defining Training and Performance Caliber: A Participant Classification Framework. Int J Sports Physiol Perform. 2022 Feb 1;17(2):317-331. doi: 10.1123/ijspp.2021-0451. Epub 2022 Dec 29. PMID: 34965513.

Authors: We appreciate the reviewer’s advice. This was done as advised.

L60 – Could rework the short sentence here in to the previous sentence.

Authors: Edited as advised.

L62-63 – For readers interpretation, could you identify an average of how many strokes to cover the 25m? This would give a better indication of the relative proportion 3 strokes have on the swim.

Authors: We understand the reviewer’s comment. This was done to avoid the wall push-off performed by the swimmers. After the 10th meter mark, swimmers are already animated by their so called “clean swim”, i.e., without any external advantage (start by the block or wall push-off). That is, researchers measure the swimmers' swim speed in the intermediate section of the swimming pool. This is a standard protocol in both experimental (Silva, A. F., Figueiredo, P., Ribeiro, J., Alves, F., Vilas-Boas, J. P., Seifert, L., & Fernandes, R. J. (2019). Integrated analysis of young swimmers’ sprint performance. Motor control, 23(3), 354-364; Barbosa, T. M., Bartolomeu, R., Morais, J. E., & Costa, M. J. (2019). Skillful swimming in age-groups is determined by anthropometrics, biomechanics and energetics. Frontiers in physiology, 10, 73) and observational studies in race analysis (Morais, J. E., Marinho, D. A., Arellano, R., & Barbosa, T. M. (2019). Start and turn performances of elite sprinters at the 2016 European Championships in swimming. Sports biomechanics, 18(1), 100-114).

In this section, swimmers may perform more complete stroke cycles, but maybe not all of them can perform five or six. Thus, to standardize this (i.e., use the same number of cycles in all swimmers), we are sure that all swimmers complete three stroke cycles. As mentioned above, this is a standard procedure in swimming studies including our research group.

L64 – Has anyone identified the reliability of the SpeedRT? Can you reference a source or any pilot data you have here?

Authors: We appreciate the reviewer’s comment. A reference about this topic was added as advised (Morouço, P., Lima, A. B., Semblano, P., Fernandes, D., Gonçalves, P., Sousa, F., ... & Vilas-Boas, J. P. (2006). Validation of a cable speedometer for butterfly evaluation. Revista Portuguesa De Ciências do Desporto, 236-239).

Discussion

L152 – A study cannot aim it is inanimate, please change accordingly.

Authors: We appreciate the reviewer’s advice. Edited accordingly.

L171 – Please add the reference after Lopes for an in-text citation.

Authors: Done as advised.

Reviewer 4 Report

Comments and Suggestions for Authors

The authors must be commended for carrying out a study regarding the assessment of swimming fluctuation using different methods. This topic is novel, the research methodology used in the study is appropriate, and the manuscript is written with good clarity. However, some issues need to be taken into consideration. Please find my specific comments below

Abstract

Please emphasize the statistical analysis used in the study.

Introduction

I suggest spreading the introduction section with details about the results of previous studies.

I suggest adding a study hypothesis.

Methods

Sample: Why did you choose this specific sample size? Why did you decide to conduct the study on young swimmers? Why is the participants' age different according to the group? Please elaborate.

Sample: I strongly suggest explaining in more detail the training regime of the participants.

Swimming speed and speed fluctuation, lines 62-63: ’’ Three consecutive stroke cycles between the 10th and 20th meters were analyzed. The average of the three-stroke cycles was used for analysis.’’ Is this measurement procedure based on some previous studies? Please elaborate.

Results

I strongly suggest presenting the results as p = 0.003 (e.g.) and not as p < 0.05.

Discussion

First paragraph, second, third and fourth sentence: At the end of these sentences, please emphasize a place in the results where the reader can see these statements: (Table 2, Figure 1...).

Add a study limitation at the end of the discussion part.

I suggest adding a conclusion section.

Author Response

REVIEWER #4

The authors must be commended for carrying out a study regarding the assessment of swimming fluctuation using different methods. This topic is novel, the research methodology used in the study is appropriate, and the manuscript is written with good clarity. However, some issues need to be taken into consideration. Please find my specific comments below

Authors: Thank you very much for the time you spent and your constructive feedback on this manuscript. We have made every effort to take on board your recommendations and comments. We hope this 2nd revised version and the responses to the comments (kindly refer to our replies below) will meet your requirements. Please note that all new changes in the revised manuscript are highlighted in yellow.

Abstract

Please emphasize the statistical analysis used in the study.

Authors: Done as advised.

Introduction

I suggest spreading the introduction section with details about the results of previous studies.

Authors: We appreciate the reviewer’s advice. The intro was elaborated as advised.

I suggest adding a study hypothesis.

Authors: Added as advised.

Methods

Sample: Why did you choose this specific sample size? Why did you decide to conduct the study on young swimmers? Why is the participants' age different according to the group? Please elaborate.

Authors: We understand the reviewer’s comment. However, please note that the sample here (young or adult swimmers) is not an issue. The question here is the speed fluctuations. Based on previous research, the research gap to be overcome is if classical statistics with discrete variables deliver the same outputs as more sensitive and accurate approaches (such as SPM). This is why we chose a group that has a completely different speed fluctuation and two with similar ones to get deeper insights about hypothetical differences. Nonetheless, we added the sample size topic in the limitations as also indicated by another reviewer.  

Sample: I strongly suggest explaining in more detail the training regime of the participants.

Authors: We added some information about this topic as advised.

Swimming speed and speed fluctuation, lines 62-63: ’’ Three consecutive stroke cycles between the 10th and 20th meters were analyzed. The average of the three-stroke cycles was used for analysis.’’ Is this measurement procedure based on some previous studies? Please elaborate.

Authors: We understand the reviewer’s comment. This was done to avoid the wall push-off performed by the swimmers. After the 10th meter mark, swimmers are already animated by their so called “clean swim”, i.e., without any external advantage (start by the block or wall push-off). That is, researchers measure the swimmers' swim speed in the intermediate section of the swimming pool. This is a standard protocol in both experimental (Silva, A. F., Figueiredo, P., Ribeiro, J., Alves, F., Vilas-Boas, J. P., Seifert, L., & Fernandes, R. J. (2019). Integrated analysis of young swimmers’ sprint performance. Motor control, 23(3), 354-364; Barbosa, T. M., Bartolomeu, R., Morais, J. E., & Costa, M. J. (2019). Skillful swimming in age-groups is determined by anthropometrics, biomechanics and energetics. Frontiers in physiology, 10, 73) and observational studies in race analysis (Morais, J. E., Marinho, D. A., Arellano, R., & Barbosa, T. M. (2019). Start and turn performances of elite sprinters at the 2016 European Championships in swimming. Sports biomechanics, 18(1), 100-114).

In this section, swimmers may perform more complete stroke cycles, but maybe not all of them can perform five or six. Thus, to standardize this (i.e., use the same number of cycles in all swimmers), we are sure that all swimmers complete three stroke cycles. As mentioned above, this is a standard procedure in swimming studies including our research group.

Results

I strongly suggest presenting the results as p = 0.003 (e.g.) and not as p < 0.05.

Authors: We added this information in table 2. Please note that in SPM data (Figure 1, panels B, C, D), this procedure doesn’t present the specific p-value. Grey areas depict differences when p < 0.017. This was added in this new version.

Discussion

First paragraph, second, third and fourth sentence: At the end of these sentences, please emphasize a place in the results where the reader can see these statements: (Table 2, Figure 1...).

Authors: Added as advised.

Add a study limitation at the end of the discussion part.

Authors: Added as advised.

I suggest adding a conclusion section.

Authors: Added as advised.